# SUSTAINBENCH: Benchmarks for Monitoring the Sustainable Development Goals with Machine Learning

**Christopher Yeh**\*
Caltech

**Chenlin Meng**\*
Stanford

**Sherrie Wang**\*
UC Berkeley

**Anne Driscoll**†
Stanford

**Erik Rozi**†
Stanford

**Patrick Liu**†
Stanford

**Jihyeon Lee**†
Stanford

**Marshall Burke**
Stanford

**David Lobell**
Stanford

**Stefano Ermon**
Stanford

## Abstract

Progress toward the United Nations Sustainable Development Goals (SDGs) has been hindered by a lack of data on key environmental and socioeconomic indicators, which historically have come from ground surveys with sparse temporal and spatial coverage. Recent advances in machine learning have made it possible to utilize abundant, frequently-updated, and globally available data, such as from satellites or social media, to provide insights into progress toward SDGs. Despite promising early results, approaches to using such data for SDG measurement thus far have largely evaluated on different datasets or used inconsistent evaluation metrics, making it hard to understand whether performance is improving and where additional research would be most fruitful. Furthermore, processing satellite and ground survey data requires domain knowledge that many in the machine learning community lack. In this paper, we introduce SUSTAINBENCH, a collection of 15 benchmark tasks across 7 SDGs, including tasks related to economic development, agriculture, health, education, water and sanitation, climate action, and life on land. Datasets for 11 of the 15 tasks are released publicly for the first time. Our goals for SUSTAINBENCH are to (1) lower the barriers to entry for the machine learning community to contribute to measuring and achieving the SDGs; (2) provide standard benchmarks for evaluating machine learning models on tasks across a variety of SDGs; and (3) encourage the development of novel machine learning methods where improved model performance facilitates progress towards the SDGs.

## 1 Introduction

In 2015, the United Nations (UN) proposed 17 Sustainable Development Goals (SDGs) to be achieved by 2030, for promoting prosperity while protecting the planet [2]. The SDGs span social, economic, and environmental spheres, ranging from ending poverty to achieving gender equality to combating climate change (see Table A1). Progress toward SDGs is traditionally monitored through statistics collected by civil registrations, population-based surveys and censuses. However, such data collection is expensive and requires adequate statistical capacity, and many countries go decades between making ground measurements on key SDG indicators [20]. Only roughly half of SDG indicators have regular data from more than half of the world's countries [94]. These data gaps severely limit the ability of the international community to track progress toward the SDGs.

---

\*Joint first authors.
†Joint second authors.

35th Conference on Neural Information Processing Systems (NeurIPS 2021) Track on Datasets and Benchmarks.

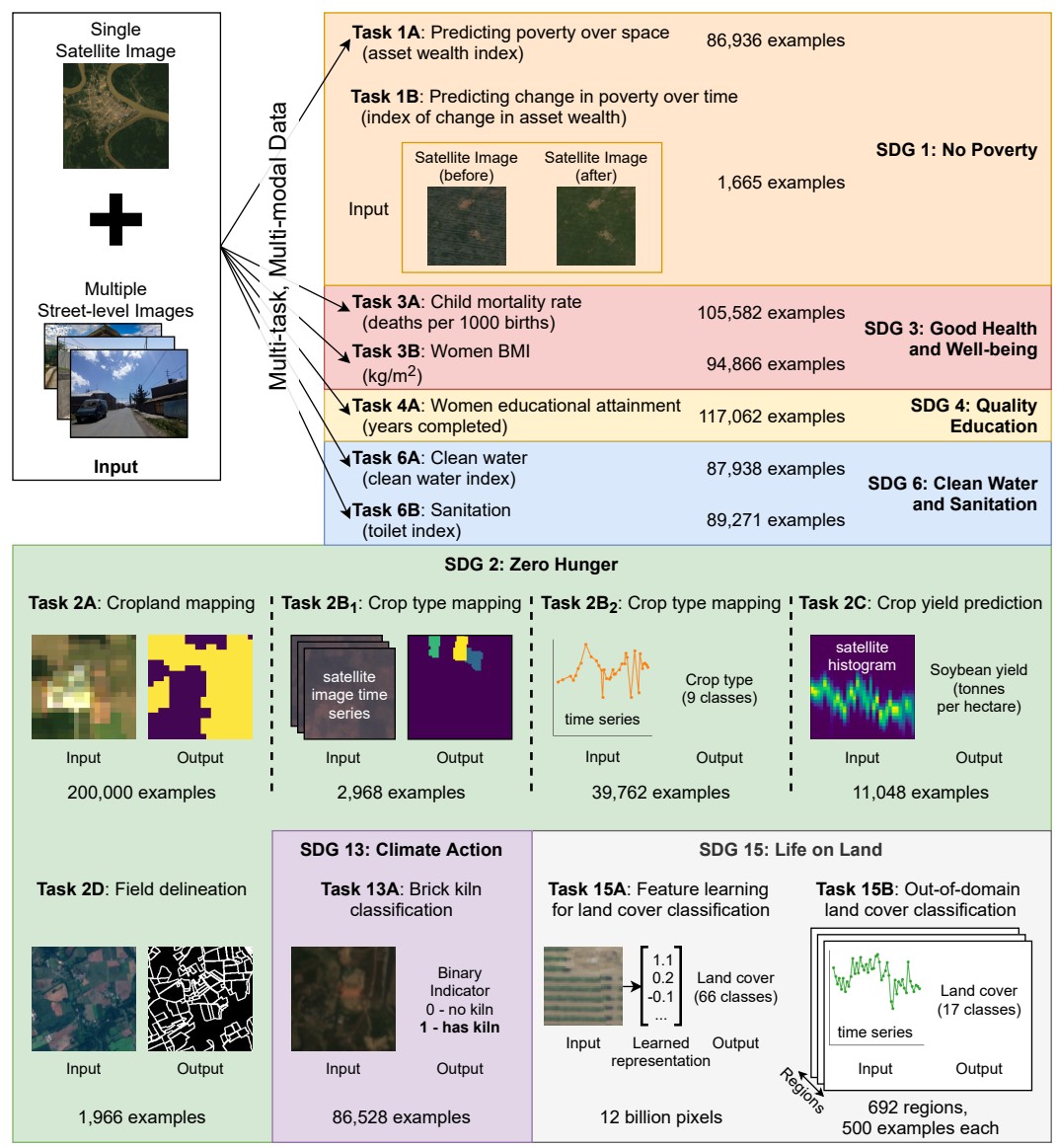

Figure 1: Datasets and tasks included in SUSTAINBENCHranging from poverty prediction to land cover classification (described in Section 3 with additional details in Appendix D). *Data for 11 out of 15 tasks are publicly released for the first time.*

Advances in machine learning (ML) have shown promise in helping plug these data gaps, demonstrating how sparse ground data can be combined with abundant, cheap and frequently updated sources of novel sensor data to measure a range of SDG-related outcomes [70, 20]. For instance, data from satellite imagery, social media posts, and/or mobile phone activity can predict poverty [15, 52, 109], annual land cover [35, 18], deforestation [42, 50], agricultural cropping patterns [69, 103], crop yields [11, 110], and the location and impact of natural disasters [25, 92]. As a timely example of real-world impact, the governments of Bangladesh, Mozambique, Nigeria, Togo, and Uganda used ML-based poverty and cropland maps generated from satellite imagery or phone records to target economic aid to their most vulnerable populations during the COVID-19 pandemic [14, 38, 56, 66]. Other recent work demonstrates using ML-based poverty maps to measure the effectiveness of large-scale infrastructure investments [78].

But further methodological progress on the "big data approach" to monitoring SDGs is hindered by a number of key challenges. First, downloading and working with both novel input data (*e.g.*, from satellites) and ground-based household surveys requires domain knowledge that many in the ML community lack. Second, existing approaches have been evaluated on different datasets, data splits,

or evaluation metrics, making it hard to understand whether performance is improving and where additional research would be most fruitful [20]. This is in stark contrast to canonical ML datasets like MNIST, CIFAR-10 [60], and ImageNet [81] that have standardized inputs, outputs, and evaluation criteria and have therefore facilitated remarkable algorithmic advances [43, 28, 57, 44, 47]. Third, methods used so far are often adapted from methods originally designed for canonical deep learning datasets (*e.g.*, ImageNet). However, the datasets and tasks relevant to SDGs are unique enough to merit their own methodology. For example, gaps in monitoring SDGs are widest in low-income countries, where only sparse ground labels are available to train or validate predictive models.

To facilitate methodological progress, this paper presents SUSTAINBENCH, a compilation of datasets and benchmarks for monitoring the SDGs with machine learning. Our goals are to

1. lower the barriers to entry by supplying high-quality domain-specific datasets in development economics and environmental science,
2. provide benchmarks to standardize evaluation on tasks related to SDG monitoring, and
3. encourage the ML community to evaluate and develop novel methods on problems of global significance where improved model performance facilitates progress towards SDGs.

In SUSTAINBENCH, we curate a suite of 15 benchmark tasks across 7 SDGs where we have relatively high-quality ground truth labels: No Poverty (SDG 1), Zero Hunger (SDG 2), Good Health and Well-being (SDG 3), Quality Education (SDG 4), Clean Water and Sanitation (SDG 6), Climate Action (SDG 13), and Life on Land (SDG 15). Figure 1 summarizes the datasets in SUSTAINBENCH. Although results for some tasks have been published previously, *data for 11 of the 15 tasks are being made public for the first time*. We provide baseline models for each task and a public leaderboard[3].

To our knowledge, this is the first set of large-scale cross-domain datasets targeted at SDG monitoring compiled with standardized data splits to enable benchmarking. SUSTAINBENCH is not only valuable to improving sustainability measurements but also offers tasks for ML challenges, allowing for the development of self-supervised learning (Section 3.7), meta-learning (Section 3.7), and multi-modal/multi-task learning methods (Sections 3.1 and 3.3 to 3.5) on real-world datasets.

In the remainder of this paper, Section 2 surveys related datasets; Section 3 introduces the SDGs and datasets covered by SUSTAINBENCH; Section 4 summarizes state-of-the-art models on each dataset and where methodological advances are needed; and Section 5 highlights the impact, limitations, and future directions of this work. The Appendix includes detailed information about the inputs, labels, and tasks for each dataset.

## 2   Related Work

Our work builds on a growing body of research that seeks to measure SDG-relevant indicators, including those cited above. These individual studies typically focus on only one SDG-related task, but even within a specific SDG domain (*e.g.*, poverty prediction), most tasks lack standardized datasets with clear replicate-able benchmarks [20]. In comparison, SUSTAINBENCH is a compilation of datasets that covers 7 SDGs and provides 15 standardized, replicate-able tasks with established benchmarks. Table 1 compares SUSTAINBENCH against existing datasets that pertain to SDGs, are publicly available, provide ML-friendly inputs/outputs, and specify standardized evaluation metrics.

Perhaps the most closely-related benchmark dataset is WILDS [59], which provides a comprehensive benchmark for distribution shifts in real-world applications. However, WILDS is not focused on SDGs, and although it includes a poverty mapping task, our poverty dataset covers $5\times$ more countries.

There also exist a number of datasets for performing satellite or aerial imagery tasks related to the SDGs [23, 86, 89, 108, 96, 62, 41, 4, 26, 96] which share similarities with the inputs of SUSTAIN-BENCH on certain benchmarks. For example, [86] compiled imagery from the Sentinel-1/2 satellites, which we also use for SDG monitoring tasks, and the Radiant Earth Foundation has compiled datasets for crop type mapping [77], a task we also include. However, SUSTAINBENCH's goal is to provide a broader view of what ML can do for SDG monitoring; it is differentiated in its focus on multiple SDGs, multiple inputs, and on low-income regions in particular. For tasks where existing datasets are abundant (*e.g.*, cropland and land cover classification), SUSTAINBENCH has tasks that address

---

[3]https://sustainlab-group.github.io/sustainbench/leaderboard

Table 1: A comparison of SUSTAINBENCH with related datasets and benchmarks. A dataset is only included if it is relevant for an SDG, is publicly available, provides both inputs and outputs in ML-friendly formats, defines train/test sets, and standardizes evaluation metrics.

| Name | Purpose | Geography | Time | Inputs | Relevant for SDGs | | | | | | | | |
|---|---|---|---|---|---|---|---|---|---|---|---|---|---|
| | | | | | 1 | 2 | 3 | 4 | 6 | 11 | 13 | 14 | 15 |
| SUSTAINBENCH | SDG monitoring | 1-105 countries/task (119 total) | 1-24 years/task in 1996-2019 | Sat. images, street-level images, and/or time series | ✓ | ✓ | ✓ | ✓ | ✓ | | ✓ | | ✓ |
| Yeh *et al.* / WILDS [109, 59] | Poverty mapping | 23 countries | 2009-16 | Sat. images | ✓ | | | | | | | | |
| Radiant MLHub [77] | Crop type mapping | 8 countries | 1-3 years/task in 2015-21 | Sat. time series or drone images | | ✓ | | | | | | | |
| SpaceNet [96] | Building & road detection | 10+ cities | Unknown | Sat. images & time series | | | | | | ✓ | | | |
| DeepGlobe [26] | Building & road detection, land cover classification | 3 countries, 4 cities | Unknown | Sat. images | | | | | | ✓ | | | ✓ |
| fMoW / WILDS [23, 59] | Object detection | 207 countries | 2002-17 | Sat. images | | | | | | ✓ | | | |
| xView [62] | Object classification | 30+ countries | Unknown | Sat. images | | | | | | ✓ | | | |
| xBD (xView2) [41] | Disaster damage assessment | 10 countries | 2011-19 | Sat. images | | | | | | ✓ | | | |
| xView3 [4] | Illegal fishing detection | Oceans | Unknown | Sat. images | | | | | | | | ✓ | |
| BigEarthNet [89] | Land cover classification | 10 countries in Europe | 2017-18 | Sat. images | | | | | | | | | ✓ |
| ForestNet [50] | Deforestation drivers | Indonesia | 2001-16 | Environ. data & sat. images | | | | | | | ✓ | | ✓ |
| iWildCam2020 / WILDS [13, 59] | Wildlife monitoring | 12 countries | 2013-15 | Camera trap images | | | | | | | | | ✓ |

remaining challenges in the domain (*e.g.*, learning from weak labels, sharing knowledge across the globe). Appendix D provides task-by-task comparisons of SUSTAINBENCH datasets with prior work.

## 3   SUSTAINBENCH Datasets and Tasks

In this section, we introduce the SUSTAINBENCH datasets and provide background on the SDGs that they help monitor. Seven SDGs are currently covered: No Poverty (SDG 1), Zero Hunger (SDG 2), Good Health and Well-being (SDG 3), Quality Education (SDG 4), Clean Water and Sanitation (SDG 6), Climate Action (SDG 13), and Life on Land (SDG 15). We describe how progress toward each goal is traditionally monitored, the gaps that currently exist in monitoring, and how certain indicators can be monitored using non-traditional datasets instead. Figure 1 summarizes the SDG, inputs, outputs, tasks, and original reference of each dataset, and Figures 2 and A1 visualize how many SDG indicators are covered by SUSTAINBENCH in each country. All of the datasets are easily downloaded via a Python package that integrates with the PyTorch ML framework [75].

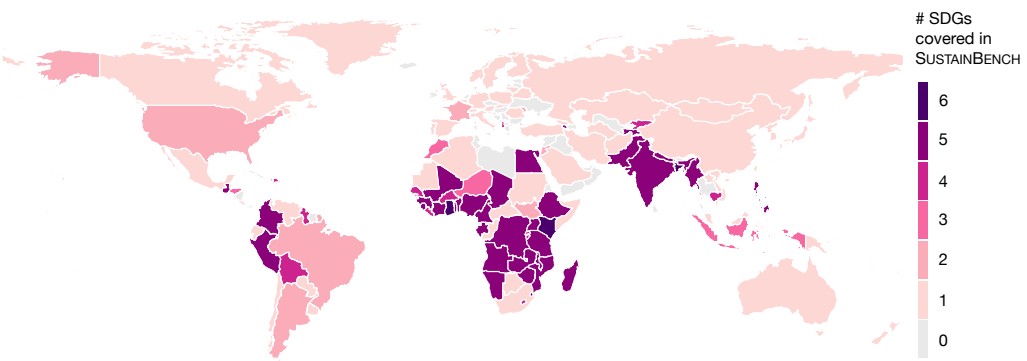

Figure 2: A map of how many SDGs are covered in SUSTAINBENCH for every country. SUSTAIN-BENCH has global coverage with an emphasis on low-income countries. In total, 119 countries have at least one task in SUSTAINBENCH.

## 3.1 No Poverty (SDG 1)

Despite decades of declining poverty rates, an estimated 8.4% of the global population remains in extreme poverty as of 2019, and progress has slowed in recent years [93]. But data on poverty remain surprisingly sparse, hampering efforts at monitoring local progress, targeting aid to those who need it, and evaluating the effectiveness of antipoverty programs [20]. In most African countries, for example, nationally representative consumption or asset wealth surveys, the key source of internationally comparable poverty measurements, are only available once every four years or less [109].

For SUSTAINBENCH, we processed survey data from two international household survey programs: Demographic and Health Surveys (DHS) [48] and the Living Standards Measurement Study (LSMS). Both constitute nationally representative household-level data on assets, housing conditions, and education levels, among other attributes. Notably, only LSMS data form a panel—*i.e.*, the same households are surveyed over time, facilitating comparison over time. Using a a principal components analysis (PCA) approach [31, 85], we summarize the survey data into a single scalar asset wealth index per "cluster," which roughly corresponds to a village or local community. We refer to cluster-level wealth (or its absence) as "poverty". Previous research has shown that widely-available imagery sources including satellite imagery [52, 109] and crowd-sourced street-level imagery [64] can be effective for predicting cluster-level asset wealth when used as inputs in deep learning models.

SUSTAINBENCH includes two regression tasks for poverty prediction at the cluster level, both using imagery inputs to estimate an asset wealth index. The first task (Section 3.1.1) predicts poverty over space, and the second task (Section 3.1.2) predicts poverty changes over time.

### 3.1.1 Poverty Prediction Over Space

The poverty prediction over space task involves predicting a cluster-level asset wealth index which represents the "static" asset wealth of a cluster at a given point in time. For this task, the labels and inputs are created in a similar manner as in [109], but with about 5× as many examples.

**Dataset** Following techniques developed in previous works [52, 109], we assembled asset wealth data for 2,079,036 households living in 86,936 clusters across 48 countries, drawn from DHS surveys conducted between 1996 and 2019, computing a cluster-level asset wealth index as described above. We provide satellite and street-level imagery inputs, gathered and processed according to established procedures [109, 64]. The 255×255×8px satellite images have 7 multispectral bands from Landsat daytime satellites and 1 nightlights band from either the DMSP or VIIRS satellites. The images are rescaled to a resolution of 30m/px and are geographically centered around each surveyed cluster's geocoordinates. Geocoordinates in the public survey data are "jittered" by up to 10km from the true locations to protect the privacy of surveyed households [19]. For each cluster location, we also retrieved up to 300 crowd-sourced, street-level imagery from Mapillary. We evaluate model performance using the squared Pearson correlation coefficient ($r^2$) between predicted and observed values of the asset wealth index on held-out test countries. Appendix D.1 has more dataset details.

### 3.1.2 Poverty Prediction Over Time

For predicting temporal changes in poverty, we construct a PCA-based index of changes in asset ownership using LSMS data. For this task, the labels and inputs provided are similar to [109], with small improvements in image and label quality.

**Dataset** We provide labels for 1,665 instances of cluster-level asset wealth change from 1,287 clusters in 5 African countries. We use the same satellite imagery sources from the previous poverty prediction task. In this task, however, for each cluster we provide images from the two points in time (before and after) used to compute the difference in asset ownership, instead of only from a single point in time. Because street-level images were only available for ∼1% of clusters, we do not provide them for this task. We evaluate model performance using the squared Pearson correlation coefficient ($r^2$) on predictions and labels in held-out cluster locations. Appendix D.2 has more dataset details.

## 3.2 Zero Hunger (SDG 2)

The number of people who suffer from hunger has risen since 2015, with 690 million or 9% of the world's population affected by chronic hunger [93]. At the same time, 40% of habitable land on Earth is already devoted to agricultural activities, making agriculture by far the largest human impact on

the natural landscape [5]. The second SDG is to "end hunger, achieve food security and improved nutrition, and promote sustainable agriculture." In addition to ending hunger and malnutrition in all forms, the targets under SDG 2 include doubling the productivity of small-scale food producers and promoting sustainable food production [93]. While traditionally data on agricultural practices and farm productivity are obtained via farm surveys, such data are rare and often of low quality [20]. Satellite imagery offers the opportunity to monitor agriculture more cheaply and more accurately, by mapping cropland, crop types, crop yields, field boundaries, and agricultural practices like cover cropping and conservation tillage. We discuss the SUSTAINBENCH datasets for SDG 2 below.

### 3.2.1 Cropland mapping with weak labels

One indicator for SDG 2 is the proportion of agricultural area under productive and sustainable agriculture [93]. Existing state-of-the-art datasets on land cover [18, 35] are derived from satellite time series and include a cropland class. However, the maps are known to have large errors in regions of the world like Sub-Saharan Africa where ground labels are sparse [56]. Therefore, while mapping cropland is largely a solved problem in settings with ample labels, devising methods to efficiently generate georeferenced labels and accurately map cropland in low-resource regions remains an important and challenging research direction.

**Dataset** We release a dataset for performing weakly supervised cropland classification in the U.S. using data from [102], which has not been released previously. While densely segmented labels are time-consuming and infeasible to generate for a large region like Africa, pixel-level and image-level labels are easier to create. The inputs are image tiles taken by the Landsat satellites and composited over the 2017 growing season, and the labels are either binary {cropland, not cropland} at single pixels or {$\geq 50\%$ cropland, $< 50\%$ cropland} for the entire image. Labels are generated from a high-quality USDA dataset on land cover [69]. Train, validation, and test sets are split along geographic blocks, and we evaluate models by overall accuracy and F1-score. We also encourage the use of semi-supervised and active learning methods to relieve the labeling burden needed to map cropland.

### 3.2.2 Crop type mapping in Sub-Saharan Africa

Spatially disaggregated crop type maps are needed to assess agricultural diversity and estimate yields. In high-income countries across North America and Europe, crop type maps are produced annually by departments of agriculture using farm surveys and satellite imagery [69]. However, no such maps are regularly available for middle- and low-income countries. Mapping crop types in the Global South faces challenges of irregularly shaped fields, small fields, intercropping, sparse ground truth labels, and highly heterogeneous landscapes [83]. We release two crop type datasets in Sub-Saharan Africa and point the reader to additional datasets hosted by the Radiant Earth Foundation [77] (Table 1). We recommend that ML researchers use all available datasets to ensure model generalizability.

**Dataset #1** We re-release the dataset from [83] in Ghana and South Sudan in a format more familiar to the ML community. The inputs are growing season time series of imagery from three satellites (Sentinel-1, Sentinel-2, and PlanetScope) in 2016 and 2017, and the outputs are semantic segmentation of crop types. Ghana samples are labeled for maize, groundnut, rice, and soybean, while South Sudan samples are labeled for maize, groundnut, rice, and sorghum. We use the same train, validation, and test sets as [83], which preserve relative percentages of crop types across the splits. We evaluate models using overall accuracy and macro F1-score.

**Dataset #2** We release the dataset used in [58] and [54] to map crop types in three regions of Kenya. Since the timing of growth and spectral signature are two main ways to distinguish crop types, the inputs are annual time series from the Sentinel-2 multi-spectral satellite. The outputs are crop types (9 possible classes). There are a total of 39,762 pixels belonging to 5,746 fields. The training, validation, and test sets are split along region rather than by field in order to develop models that generalize across geography. Our evaluation metrics are overall accuracy and macro-F1 score.

### 3.2.3 Crop yield prediction in North and South America

In order to double the productivity (or yield) of smallholder farms, we first have to measure it, and accurate local-level yield measurements are exceedingly rare in most of the world. In SUSTAINBENCH, we release county-level yields collected from various government databases; these can still aid in forecasting production, evaluating agricultural policy, and assessing the effects of climate change.

**Dataset** Our dataset is based on the datasets used in [110] and [101]. We release county-level yields for 857 counties in the U.S., 135 in Argentina, and 32 in Brazil for the years 2005-16. The inputs are spectral band and temperature histograms over each county for the harvest season from the MODIS satellite. The ground truth labels are the regional soybean yield per harvest, in metric tonnes per cultivated hectare, retrieved from government data. See Appendix D.6 for more details. Models are evaluated using root mean squared error (RMSE) and $R^2$ of predictions with the ground truth. The imbalance of data by country motivates the use of transfer learning approaches.

### 3.2.4 Field delineation in France

Since agricultural practices are usually implemented on the level of an entire field, field boundaries can help reduce noise and improve performance when mapping crop types and yields. Furthermore, field boundaries are a prerequisite for today's digital agriculture services that help farmers optimize yields and profits [98]. Statistics that can be derived from field delineation, such as the size and distribution of crop fields, have also been used to study productivity [21, 27], mechanization [61], and biodiversity [37]. Field boundary datasets are rare and only sparsely labeled in low-income regions, so we release a large dataset from France to aid in model development.

**Dataset** We re-release the dataset introduced in Aung et al. 9. The dataset consists of Sentinel-2 satellite imagery in France over 3 time ranges: January-March, April-June, and July-September in 2017. The image has resolution 224×224 corresponding to a 2.24km×2.24km area on the ground. Each satellite image comes along with the corresponding binary masks of boundaries and areas of farm parcels. The dataset consists of a total of 1966 samples. We use a different data split from [9] to remove overlapping between the train, validation and test split. Following [9], we use the Dice score between the ground truth boundaries and predicted boundaries as the performance metric.

## 3.3 Good Health and Well-being (SDG 3)

Despite significant progress on improving global health outcomes (*e.g.*, halving child mortality rates since 2000 [93]), the lack of local-level measurements in many developing countries continues to constrain the monitoring, targeting, and evaluation of health interventions. We examine two health indicators: female body mass index (BMI), a key input to understanding both food insecurity and obesity; and child mortality rate (deaths under age 5), an official SDG 3 indicator considered to be a summary measure of a society's health. Previous works have demonstrated using satellite imagery [67] or street-level Mapillary imagery inputs [64] for predicting BMI. While we are unaware of any prior works using such imagery inputs for predicting child mortality rates, "there is evidence that child mortality is connected to environmental factors such as housing quality, slum-like conditions, and neighborhood levels of vegetation" [51], which are certainly observable in imagery.

**Dataset** We provide cluster-level average labels for women's BMI and child mortality rates compiled from DHS surveys. There are 94,866 cluster-level BMI labels computed from 1,781,403 women of childbearing age (15-49), excluding pregnant women. There are 105,582 cluster-level labels for child mortality rates computed from 1,936,904 children under age 5. As in the poverty prediction over space task (Section 3.1.1), the inputs for predicting the health labels are satellite and street-level imagery, and models are evaluated using the $r^2$ metric on labels from held-out test countries.

## 3.4 Quality Education (SDG 4)

SDG 4 includes targets that by 2030, all children and adults "complete free, equitable and quality primary and secondary education". Increasing educational attainment (measured by years of schooling completed) is known to increase wealth and social mobility, and higher educational attainment in women is strongly associated with improved child nutrition and decreased child mortality [40]. Previous works have demonstrated the ability of deep learning methods to predict educational attainment from both satellite images [112] and street-level images [36, 64].

**Dataset** We provide cluster-level average years of educational attainment by women of reproductive age (15-49) compiled from same DHS surveys used for creating the asset wealth labels in the poverty prediction task. The 122,435 cluster-level labels were computed from 3,013,286 women across 56 countries. As in the poverty prediction over space task (Section 3.1.1), the inputs for predicting women educational attainment are satellite and street-level imagery, and models are evaluated using the $r^2$ metric on labels from held-out test countries.

### 3.5 Clean Water and Sanitation (SDG 6)

Clean water and sanitation are fundamental to human health, but as of 2020, two billion people globally do not have access to safe drinking water, and 2.3 billion lack a basic hand-washing facility with soap and water [84]. Access to improved sanitation and clean water is known to be associated with lower rates of child mortality [65, 33].

**Dataset**  We provide cluster-level average years of a water quality index and sanitation index compiled from same DHS surveys used for creating the asset wealth labels in the poverty prediction task. The 87,938 (water index) and 89,271 (sanitation index) cluster-level labels were computed from 2,105,026 (water index) and 2,143,329 (sanitation index) households across 49 countries. As in the poverty prediction over space task (Section 3.1.1), the inputs for predicting the water quality and sanitation indices are satellite and street-level imagery, and models are evaluated using the $r^2$ metric on labels from held-out test countries. Since SUSTAINBENCH includes labels for child mortality in many of the same clusters with sanitation index labels, we encourage researchers to take advantage of the known associations between these variables.

### 3.6 Climate Action (SDG 13)

SDG 13 aims at combating climate change and its disruptive impacts on national economies and local livelihoods [68]. Monitoring emissions and environmental regulatory compliance are key steps toward SDG 13.

#### 3.6.1 Brick kiln mapping

Brick manufacturing is a major source of carbon emissions and air pollution in South Asia, with an industry largely comprised of small-scale, informal producers. Identifying brick kilns from satellite imagery is a scalable method to improve compliance with environmental regulations and measure their impact on nearby populations. A recent study [63] trained a CNN to detect kilns and hand-validated the predictions, providing ground truth kiln locations in Bangladesh from October 2018 to May 2019.

**Dataset**  The high-resolution satellite imagery used in [63] could not be shared publicly because they were proprietary. Hence, we provide a lower resolution alternative—Sentinel-2 imagery, which is available through Google Earth Engine [39]. We retrieved $64 \times 64 \times 13$ tiles at 10m/pixel resolution from the same time period and labeled each image as not containing a brick kiln (class 0) or containing a brick kiln (class 1) based on the ground truth locations in [63]. There were 6,329 positive examples out of 374,000 examples total; we sampled 25% of the negative examples and removed null values, resulting in 67,284 negative examples. More details can be found in Appendix D.8.

### 3.7 Life on Land (SDG 15)

Human activity has altered over 75% of the earth's surface, reducing forest cover, degrading once-fertile land, and threatening an estimated 1 million animal and plant species with extinction [93]. Our understanding of land cover—*i.e.*, the physical material on the surface of the earth—and its changes is not uniform across the globe. Existing state-of-the-art land cover maps [18] are significantly more accurate in high-income regions than low-income ones, as the latter have few ground truth labels [56]. The following two datasets seek to reduce this gap via representation learning and transfer learning.

#### 3.7.1 Representation learning for land cover classification

One approach to increase the performance of land cover classification in regions with few labels is to use unsupervised or self-supervised learning to improve satellite/aerial image representations, so that downstream tasks require fewer labels to perform well.

**Dataset**  We release the high-resolution aerial imagery dataset from [53], which spans a 2500km$^2$ (12 billion pixel) area of Central Valley, CA in the U.S. The output is image-level land cover (66 classes), where labels are generated from a high-quality USDA dataset [69]. The region is divided in geographically-continuous blocks into train, validation, and test sets. The user may use the training imagery in any way to learn representations, and we provide a test set of up to 200,000 tiles (100×100px) for evaluation. The evaluation metrics are overall accuracy and macro F1-score.

Table 2: Benchmark performance on 15 tasks across 7 SDGs. See details in Appendix E. For the Model Type column, kNN = k-nearest neighbors, GP = Gaussian process. An asterisk (*) indicates a result on a similar dataset, but not the exact SUSTAINBENCH test set.

| SDG | Task | Countries | Metric | Benchmark Value | Model Type | Ref |
|---|---|---|---|---|---|---|
| No Poverty | Poverty prediction over space | 48 countries | $r^2$ | 0.63 | kNN | [109] |
| | Poverty prediction over time | 5 African countries | $r^2$ | 0.35* | ResNet-18 | [109] |
| Zero Hunger | Weakly supervised cropland classification | United States | F1 score | 0.88 (pixel label) 0.80 (image label) | U-Net | [102] |
| | Crop type classification | Ghana, South Sudan | Macro F1 | 0.57, 0.70 | LSTM | [83] |
| | | Kenya | Macro F1 | 0.30 | Random forest | [58] |
| | Crop yield prediction | United States | RMSE | 0.37 t/ha | CNN+GP | [110] |
| | | Argentina, Brazil | | 0.62 t/ha, 0.42 t/ha | LSTM | [101] |
| | Field delineation | France | Dice score | 0.61 | U-Net | [9] |
| | | | | 0.87 | FracTAL Res-UNet | [99] |
| Good Health & Well-Being | Child mortality rate | 56 countries | $r^2$ | 0.01 | kNN | – |
| | Women BMI | 53 countries | $r^2$ | 0.42 | kNN | – |
| Quality Education | Women education | 53 countries | $r^2$ | 0.26 | kNN | – |
| Clean Water and Sanitation | Water index | 49 countries | $r^2$ | 0.40 | kNN | – |
| | Sanitation index | 49 countries | $r^2$ | 0.36 | kNN | – |
| Climate Action | Brick kiln detection | Bangladesh | Accuracy | 0.94* | ResNet-50 | [63] |
| Life on Land | Representation learning for land cover | United States | Accuracy | 0.55 ($n = 1,000$) 0.58 ($n = 10,000$) | Tile2Vec with ResNet-50 | [53] |
| | Out-of-domain land cover classification | Global | Kappa | 0.32 (1-shot, 2-way) | MAML with shallow 1D CNN | [104] |

### 3.7.2 Out-of-domain land cover classification

A second strategy for increasing performance in label-scarce regions is to transfer knowledge learned from classifying land cover in high-income regions to low-income ones.

**Dataset** We release the global dataset of satellite time series from [104]. The dataset samples 692 regions of size 10km × 10km around the globe; for each region, 500 latitude/longitude coordinates are sampled. The input is time series from the MODIS satellite over the course of a year, and the output is land cover type (17 possible classes). Users have the option of splitting regions into train, validation, and test sets at random or by continent. The evaluation metrics are overall accuracy, F1-score, and kappa score. The results from [104] are reported with all regions from Africa as the test set, but the user can choose to hold out other continents, for which the label quality will be higher.

## 4 Results for Baseline Models

SUSTAINBENCHprovides a benchmark and public leaderboard website for the datasets described in Section 3. Each dataset has standard train-test splits with well-defined performance metrics detailed in Appendix E. We also welcome community submissions using additional data sources beyond what is provided in SUSTAINBENCH, such as for pre-training or regularization. Table 2 summarizes the baseline models and results. Code to reproduce our baseline models is available on GitHub[4].

Here, we highlight some main takeaways from our baseline models. First, there is significant room for improvement for models that can take advantage of multi-modal inputs. Specifically, our baseline model for the DHS survey-based tasks only uses the satellite imagery inputs, and its poor performance on predicting child mortality and women educational attainment demonstrates the need to leverage additional data sources, such as the street-level imagery we provide. Second, ML model development can lead to significant gains in performance for SDG-related tasks. While the original paper that compiled SUSTAINBENCH's field delineation dataset achieved a Dice score of 0.61 with a standard U-Net [9], we applied a new attention-based CNN developed specifically for field delineation [99] and achieved a 0.87 Dice score. For more task-specific discussions, please see Appendix E.

## 5 Impact, Limitations, and Future Work

This paper introduces SUSTAINBENCH, which, to the best of our knowledge, is the largest compilation to date of datasets and benchmarks for monitoring the SDGs with machine learning (ML). The SDGs

---

[4]https://github.com/sustainlab-group/sustainbench/

are arguably the most urgent challenges the world faces today, and it is important that the ML community contribute to solving these global issues. As progress towards SDGs is often hindered by a lack of ground survey data especially in low-income countries, ML algorithms designed for monitoring SDGs are important for leveraging non-traditional data sources that are cheap, globally available, and frequently-updated to fill in data gaps. ML-based estimates provide policymakers from governments and aid organizations with more frequent and comprehensive insights [109, 20, 52].

The tasks defined in SUSTAINBENCH can directly translate into real-world impact. For example, during the COVID-19 pandemic, the government of Togo collaborated with researchers to use satellite imagery, phone data, and ML to map poverty [14] and cropland [56] in order to target cash payments to the jobless. Recent work in Uganda demonstrates how ML-based poverty maps can be used to measure the effectiveness of large-scale infrastructure investments [78]. ML-based analyses of satellite images in Kenya (using the labels described in Section 3.2.2) were recently used to identify soil nitrogen deficiency as the limiting factor in maize yields, thereby facilitating targeted agriculture intervention [54]. And as a last example, the development of a new attention-based neural network architecture enabled the delineation of 1.7 million fields in Australia from satellite imagery [99]. These field boundaries have been productized and facilitate the adoption of digital agriculture, which can improve yields while minimizing environmental pollution [24].

Although ML approaches have demonstrated value on a variety of tasks related to SDGs [109, 20, 64, 53, 52, 101, 103], the "big data approach" has its limits. ML models may not completely replace ground surveys. Imperfect predictions from ML models may introduce biases that propagate through downstream policy decisions, leading to negative societal impacts. The use of survey data, high resolution remote sensing images, and street-level images may also raise privacy concerns, despite efforts to protect individual privacy. We refer the reader to Appendix F for a detailed treatment of ethical concerns in SUSTAINBENCH, including mitigation strategies we implemented. Despite these limitations, ML applications have the greatest potential for positive impact in low-income countries, where gaps in monitoring SDGs are widest due to the constant lack of survey data.

While SUSTAINBENCH is the largest SDG-focused ML dataset and benchmark to date, it is by no means complete. Field surveys are extremely costly, and labeling images for model training requires significant manual effort by experts, limiting the amount of data released in SUSTAINBENCH to quantities smaller than those of many canonical ML datasets (*e.g.*, ImageNet). In addition, many SDGs and indicators are not included in the current version. Such SDG indicators can be placed into 3 categories. First, several tasks can be included in future versions of SUSTAINBENCH by drawing on existing data. For example, measures of gender equality (SDG 5) and access to affordable and clean energy (SDG 7) already exist in the surveys used to create labels for SUSTAINBENCH tasks but will require additional processing before releasing. Recent works have also pioneered deep learning methods for identifying illegal fishing from satellite images [74] (SDG 14) and monitoring biodiversity from camera traps [13] (SDG 15). Table 1 includes a few relevant datasets from this first category. Second, some SDG indicators require additional research to discover non-traditional data modalities that can be used to monitor them. Finally, not all SDGs are measurable using ML or need improved measurement capabilities from ML models. For example, international cooperation (SDG 17) is perhaps best measured by domestic and international policies and agreements.

For the ML community, SUSTAINBENCH also provides opportunities to test state-of-the-art ML models on real-world data and develop novel algorithms. For example, the tasks based on DHS household survey data share the same inputs and thus facilitate multi-task training. In particular, we encourage researchers to take advantage of the known strong associations between asset wealth, child mortality, women's education, and sanitation labels [33, 40]. The combination of satellite and street-level imagery for these tasks also enables multi-modal representation learning. On the other hand, the land cover classification and cropland mapping tasks provide new real-world datasets for evaluating and developing self-supervised, weakly supervised, unsupervised, and meta-learning algorithms. We welcome exploration of methods beyond our provided baseline models.

Ultimately, we hope SUSTAINBENCH will lower the barrier to entry for the ML community to contribute toward monitoring SDGs and highlight challenges for ML researchers to address. In the long run, we plan to continue expanding datasets and benchmarks as new data sources become available. We believe that standardized datasets and benchmarks like those in SUSTAINBENCH are imperative to both novel method development and real-world impact.

## Acknowledgments

The authors would like to thank everyone from the Stanford Sustainability and AI Lab for constructive feedback and discussion; the Mapillary team for technical support on the dataset; Rose Rustowicz for helping compile the crop type mapping dataset in Ghana and South Sudan; Anna X. Wang and Jiaxuan You for their help in making the crop yield dataset; and Han Lin Aung and Burak Uzkent for permission to release the field delineation dataset.

This work was supported by NSF awards (#1651565, #1522054), the Stanford Institute for Human-Centered AI (HAI), the Stanford King Center, the United States Agency for International Development (USAID), a Sloan Research Fellowship, and the Global Innovation Fund.

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
