# OpenReview forum: "SustainBench: Benchmarks for Monitoring the Sustainable Development Goals with Machine Learning"
_NeurIPS.cc/2021/Track/Datasets_and_Benchmarks/Round2 — NeurIPS 2021 Datasets and Benchmarks Track (Round 2)_

### Official Review · Reviewer_8Ey7 · 2021-09-11
**A lot of datasets for one paper, looks interesting, needs better presentation, what's the impact?**

**Rating:** 7
**Confidence:** 3
**Clarity:** Generally yes.

**Strengths:**

Lots of datasets, the work looks well done.  Impact demonstrated with some of the follow on work, this looks pretty nice.

**Weaknesses:**

I think the biggest weakness is that I don't know a ton about  remote sensing and don't have an intuitive idea of how great each dataset is.

**Additional Feedback:**

Thank you for incorporating feedback, I feel like this was very helpful and the result  is much clearer to me as a reader.  I'm changing my score to a 7.

**Correctness:**

There are 15 datasets here. I'm going to assume yes, but frankly, I'm not a good person to assess this.

**Documentation:**

Probably.

**Ethics:**

Ethically looks good.

**Relation To Prior Work:**

Yes

**Summary And Contributions:**

The authors are releasing ~10 novel datasets and improving upon ~5 already available ones. I think they have dev/train/test splits, which is great. That's a lot of datasets, especially since they tend to cover countries that are poorer than e.g., US and Western Europe. It  should be very interesting for the community. The value and impact of the datasets is unknown (as far as the reader can tell). They are also heavily oriented around remote sensing.

---

> ### Author Response · Authors · 2021-09-30
> **References for Response to Reviewer 8Ey7**
>
> **References:**
>
> [1] J. Blumenstock. Machine learning can help get COVID-19 aid to those who need it most. Nature, 2020.
>
> [2] H. Kerner, G. Tseng, I. Becker-Reshef, C. Nakalembe, B. Barker, B. Munshell, M. Paliyam, and M. Hosseini. Rapid response crop maps in data sparse regions, 2020.
>
> [3] Z. Jin, G. Azzari, C. You, S. Di Tommaso, S. Aston, M. Burke, and D. B. Lobell. Smallholder maize area and yield mapping at national scales with Google Earth Engine. Remote Sensing of Environment, 2019.
>
> [4] ​​F. Waldner, F. I. Diakogiannis, K. Batchelor, M. Ciccotosto-Camp, E. Cooper-Williams, C. Herrmann, G. Mata, and A. Toovey.  Detect, consolidate, delineate:  Scalable mapping of field boundaries using satellite images. Remote Sensing, 2021.
>
> [5] CSIRO. ePaddocks Australian Paddock Boundaries. URL: https://acds.csiro.au/epaddock-australian-paddock-boundaries
>
> [6] C. Yeh, A. Perez, A. Driscoll, G. Azzari, Z. Tang, D. Lobell, S. Ermon, and M. Burke. Using publicly available satellite imagery and deep learning to understand economic well-being in Africa. Nature Communications, 2020.
>
> [7] S. Wang, W. Chen, S. M. Xie, G. Azzari, and D. B. Lobell. Weakly supervised deep learning for segmentation of remote sensing imagery. Remote Sensing, 2020.
>
> [8] L. Yan and D. Roy. Conterminous United States crop field size quantification from multi-temporal Landsat data.Remote Sensing of Environment, 2016.
>
> [9] ​​T. Gebru, J. Krause, Y. Wang, D. Chen, J. Deng, E. L. Aiden, and L. Fei-Fei. Using deep learning  and  Google  Street  View  to  estimate  the  demographic  makeup  of  neighborhoods across the United States. Proceedings of the National Academy of Sciences, 2017.
>
> [10] ​​J. Lee,  D.  Grosz,  B.  Uzkent,  S.  Zeng,  M.  Burke,  D.  Lobell,  and  S.  Ermon.   Predicting Livelihood Indicators from Community-Generated Street-Level Imagery. Proceedings of the AAAI Conference on Artificial Intelligence, 2021.
>
> [11] J. Lee, N. Brooks, F. Tajwar, M. Burke, S. Ermon, D. Lobell, D. Biswas, and S. Luby. Scalable deep learning to identify brick kilns and aid regulatory capacity. Proceedings of the National Academy of Sciences, 118(17), 2021. ISSN 0027-8424. doi: 10.1073/pnas.2018863118. URL https://www.pnas.org/content/118/17/e2018863118
>
> [12] Stanford Woods Institute for the Environment. A Better Brick: Solving an Airborne Threat. URL: https://woods.stanford.edu/research/funding-opportunities/environmental-venture-projects/brick-kiln-solutions
>
> [13] S. Wang, M. Rußwurm, M. Körner, and D. B. Lobell. Meta-learning for few-shot time series classification. In IGARSS 2020 - 2020 IEEE International Geoscience and Remote Sensing Symposium, 2020
>
> [14] A. Head, M. Manguin, N. Tran, and J. E. Blumenstock. Can Human Development be Measured with Satellite Imagery? In Proceedings of the Ninth International Conference on Information and Communication Technologies and Development, pages 1–11, Lahore, Pakistan, 11 2017. ACM. ISBN 978-1-4503-5277-2. doi: 10.1145/3136560.3136576. URL http://dl.acm.529org/citation.cfm?doid=3136560.3136576.
>
> [15] A. Maharana and E. O. Nsoesie. Use of Deep Learning to Examine the Association of the Built Environment With Prevalence of Neighborhood Adult Obesity. JAMA Network Open, 1(4):e181535, 8 2018. ISSN 2574-3805. doi: 10.1001/jamanetworkopen.2018.1535. URL https://doi.org/10.1001/jamanetworkopen.2018.1535.

---

> ### Author Response · Authors · 2021-09-30
> **Response to Reviewer 8Ey7 (Part 3 of 3: Novelty of Individual Datasets)**
>
> _**Q:** Explain the novelty of individual datasets relative to prior work._
>
> **A:** Here is a list summarizing the novelty of each dataset in SustainBench. See the updated paper for longer explanations.
>
> - **Poverty prediction over space**: 1st dataset combining both satellite & street-level imagery for poverty prediction, with approx. 5x as many labels as the previous state-of-the-art dataset (~100K vs. ~20K).
> - **Poverty prediction over time**: re-release of the LSMS dataset from [6], with improvements to input and label quality. This dataset (along with original dataset in [6]) is the only publicly-available benchmark dataset that tests models on predicting changes in poverty over time.
> - **Cropland mapping**: 1st dataset for weakly supervised cropland mapping, used in [7] to demonstrate that a U-Net trained on single-pixel or image-level labels can outperform previous state-of-the-art models on cropland segmentation. The focus of this dataset is on weak labels instead of fully segmented labels because it is much cheaper and faster to generate labels at single pixels or for an entire image than segmenting all pixels of a satellite image, especially at the scale of a country or continent.
> - **Crop type mapping (Ghana / S. Sundan)**: re-release of a previous dataset with more input imagery and more ML-friendly formatting.
> - **Crop type mapping (Kenya)**: new release, one of the largest in a smallholder system (5,746 fields). One of the train/val/test split options is designed to test model generalizability across geography by splitting along geographic clusters, which no other crop type datasets do.
> - **Crop yield**: 1st release of datasets used in 2 previous works. Very few other crop yield datasets exist, because yields require expensive farm survey techniques to measure. Datasets that do contain field-level yields are privately held by researchers, government agencies, or NGOs. SustainBench’s datasets therefore provide yields at the county level. Furthermore, crop yield prediction is challenging as it requires processing a temporal sequence of satellite images. We provide ML-friendly inputs in the form of histograms of weather and satellite features over each county.
> - **Field delineation**: 1st field boundary dataset with satellite image inputs and ML-friendly outputs. Some countries in Europe (e.g. France) have released vector files of field boundaries on their government websites, but without corresponding satellite imagery inputs or raster field boundary outputs. We provide these inputs and outputs. While field segmentation datasets from the US, South Africa, and Australia were used in prior field delineation research [4,5,8], none of those datasets are publicly available. We are also currently collecting field boundaries in low-income countries, but this data will be added to SustainBench at a later date, not in time for this submission.
> - **Women BMI, women education, and water quality**: largest ML dataset for these tasks. While previous works have demonstrated the ability of ML to estimate these indicators from satellite or street-level imagery, these works have either only been tested on the U.S. [9,15] (which is not a developing country) or only on a small number of developing countries [10,14]. By providing data from over 40 countries on these tasks, SustainBench vastly exceeds the size of previous datasets, and does so while providing multi-model inputs.
> - **Child mortality**: 1st dataset to provide satellite & street-level imagery for predicting child mortality rates. Previous works found strong statistical correlations between child mortality rates and housing conditions and local vegetation, both of which are observable in our provided imagery inputs.
> - **Sanitation index**: 1st dataset for predicting cluster-level sanitation using satellite or street-level imagery.
> - **Brick kiln detection**: 1st publicly released dataset of this size and quality on detecting brick kilns across Bangladesh from satellite imagery. This dataset was manually labeled in-house by experts. Brick kiln detection is challenging because of the sparsity of kilns and lack of similar training data. Satellite-based monitoring of kilns [11] is already affecting policy developed by public health experts, industry stakeholders (e.g. kiln owners), and government agencies [12].
> - **Representation learning for land cover classification**: This dataset is designed as a benchmark to test representation learning on high resolution satellite images, just as traditional computer vision research tests representation learning on  ImageNet and Pascal VOC.
> - **Out-of-domain land cover classification**: 1st release of dataset from [13], and 1st release of any few-shot learning dataset for satellite data. We hope this dataset will become a standard for evaluating few-shot learning algorithms on real-world time series, and that new algorithms will enable knowledge sharing from high-income regions to low-income ones.

---

> ### Author Response · Authors · 2021-09-30
> **Response to Reviewer 8Ey7 (Part 2 of 3: Summary of Related Datasets)**
>
> _**Q:** Summary table of the datasets (Dataset, license, parameters, novelty relative to SOTA)_
>
> **A:** We have added a new table (Table 1) to the manuscript, which compares SustainBench to the most relevant existing datasets / benchmarks. A Markdown version of this table is reproduced below. Notably, SustainBench has the most comprehensive coverage of SDGs of any existing dataset / benchmark, with wide geographic and temporal coverage.
>
> **Table 1**: A comparison of SustainBench with related datasets and benchmarks. A dataset is only included if it is relevant for an SDG, is publicly available, provides both inputs and outputs in ML-friendly formats, defines train/test sets, and standardizes evaluation metrics.
>
> | Name | Purpose | Geography | Time | Inputs | SDG 1 | SDG 2 | SDG 3 | SDG 4 | SDG 6 | SDG 11 | SDG 13 | SDG 14 | SDG 15 |
> |:--|:--|:--|:--|:--|:--|:--|:--|:--|:--|:--|:--|:--|:--|
> | SustainBench | SDG monitoring | 1-105 countries/task (119 total) | 1-24 years/task in 1996-2019 | Sat. images, street-level images, and/or time series | Y | Y | Y | Y | Y | | Y | | Y |
> | Yeh et al. / WILDS | Poverty mapping | 23 countries | 2009-16 | Sat. images | Y |
> | Radiant MLHub | Crop type mapping | 8 countries | 1-3 years/task in 2015-21 | Sat. time series or drone images | | Y | | | | | | | |
> | SpaceNet | Building and road detection | 10+ cities | Unknown | Sat. images and time series | | | | | | Y | | | |
> | DeepGlobe | Building and road detection, land cover classification | 3 countries, 4 cities | Unknown | Sat. images | | | | | | Y | | | Y |
> | fMoW / WILDS | Object detection | 207 countries | 2002-17 | Sat. images | | | | | | Y | | | |
> | xView | Object classification | 30+ countries | Unknown | Sat. images | | | | | | Y | | | |
> | xBD (xView2) | Disaster damage assessment | 10 countries | 2011-19 | Sat. images | | | | | | Y |
> | xView3 | Illegal fishing detection | Oceans | Unknown | Sat. images | | | | | | | | Y |
> | BigEarthNet | Land cover classification | 10 countries in Europe | 2017-18 | Sat. images | | | | | | | | | Y |
> | ForestNet | Deforestation drivers | Indonesia | 2001-16 | Environ. data and sat. images | | | | | | | Y | | Y |
> | iWildCam2020 / WILDS | Wildlife monitoring | 12 countries | 2013-15 | Camera trap images | | | | | | | | | Y |
>
> Figure 1 in the manuscript provides a complete summary of the datasets included in SustainBench. If you have suggestions for how we can improve Figure 1 for clarity, please let us know, and we will gladly update Figure 1 for the camera-ready. All of the datasets are released under the same Creative Commons license that the entirety of SustainBench licensed under. In terms of novelty, the main point we’d like to emphasize is that almost all of the tasks included in SustainBench had no previous standard benchmark. Providing a standardized benchmark on these tasks is the first and foremost novel contribution of SustainBench. That said, we chose all of the datasets in SustainBench to be novel in their own right (see our response to the next question below). Due to the amount of domain knowledge required to describe some of the datasets, we find it difficult to summarize novelty in a single table, but do elaborate on each dataset’s novelty in Appendix D. If you have suggestions for a tabular summary, we are open to ideas!

---

> ### Author Response · Authors · 2021-09-30
> **Response to Reviewer 8Ey7 (Part 1 of 3: Impact of SustainBench)**
>
> Thank you for your feedback as a non-domain expert, as it is important to us that SustainBench’s impact be as clear as possible to the ML community. Please find our point-by-point response to the reviewer’s comments below. We added a [top-level comment](https://openreview.net/forum?id=5HR3vCylqD&noteId=KLX4Yr4iaoL) to summarize the major changes made in response to all reviewers’ comments. We updated the manuscript PDF; the Supplementary Material includes a “tracked changes” PDF to help reviewers identify changes.
>
> _**Q:** Elaborate on the impact of the datasets - i.e. demonstrated value and novel insights._
>
> **A:** Thank you for suggesting that we elaborate on the impact of the datasets. In Sections 1 (Introduction) and 5 (Impacts, Limitations, and Future Work), we added multiple examples of how ML-derived products have recently enabled real-world impact. While most of these examples are not the result of SustainBench itself, our goal is for SustainBench to catalyze more of these real-world impacts. To draw an analogy, ImageNet itself did not directly make self-driving cars possible, but it facilitated the development of better computer vision models which could. The examples are summarized below and further discussed in our updated paper.
>
> - *Poverty prediction and cropland segmentation*: As a timely example of real-world impact, during the COVID-19 pandemic, the governments of Togo, Uganda, and Bangladesh used ML-based poverty and cropland maps generated from satellite imagery or phone records for deciding who should receive targeted economic aid [1,2]. The poverty mapping and cropland segmentation tasks from SustainBench are relevant here.
>
> - *Crop type prediction*: ML-based analyses of satellite images in Kenya (using the same set of labels included in SustainBench for crop type prediction) were recently used to identify soil nitrogen deficiency as the limiting factor in maize yields, thereby facilitating targeted agriculture intervention [3].
>
> - *Field delineation*: Recently, the development of a new attention-based neural network architecture enabled the delineation of 1.7 million fields in Australia from satellite imagery [4]. Automated field delineation makes it easier for farmers to access field-level analytics; previously, manual boundary input was a major deterrent from adopting digital agriculture [5]. Digital agriculture can improve yields while minimizing the use of inputs like fertilizer that cause environmental pollution -- with the net effect of increasing farmer profit. This is an example where a novel deep learning architecture enabled the creation of operational products in agriculture.
>
> - *Infrastructure planning*: Researchers used CNN models trained to predict poverty in Africa to estimate the effect of electrification on local-level economic livelihood in Uganda. The CNN models were trained on an asset wealth index created very similarly to the poverty labels included in SustainBench, except on a smaller region (~25 countries vs. ~50 countries included in SustainBench). After applying causal inference methods, the researchers were able to attribute a doubling of the economic growth rate in electrified regions to the electrification itself. This ability to causally quantify the impact of electric grid infrastructure investment across an entire country is made possible by ML models and has immediate impact on policymaking around infrastructure investment.
>
> The detailed versions of these and other examples can be found in their respective Appendix sections. We also added new references to the paper to support these examples.
>
>
> _**Q:** Train models on the datasets to demonstrate impact._
>
> **A:** We provide baseline models for all of the tasks in SustainBench. We have moved the table of baseline model results into the main text (originally in the appendix) and added a paragraph that highlights key takeaways. The paragraph is reproduced below:
>
> "Here, we highlight some main takeaways from our baseline models. First, there is significant room for improvement for models that can take advantage of multi-modal inputs. Specifically, our baseline model for the DHS survey-based tasks only uses the satellite imagery inputs, and its poor performance on predicting child mortality and women's education demonstrates the need to leverage additional data sources, such as the street-level imagery we provide. Second, ML model development can lead to significant gains in performance for SDG-related tasks. While the original paper that compiled SustainBench's field delineation dataset achieved a Dice score of 0.61 with a standard U-Net, we applied a new attention-based CNN developed specifically for field delineation and achieved a 0.87 Dice score. For more task-specific discussions, please see Appendix E."

---

> > ### Comment · Reviewer_8Ey7 · 2021-10-04
> > **But better with updates**
> >
> > Thanks for taking into account  the feedback. I think I have a much better idea of what is going on now.

---

### Official Review · Reviewer_uRQg · 2021-09-20
**A comprehensive compilation of datasets that boosts ML research towards Sustainable Development Goals**

**Rating:** 7
**Confidence:** 3
**Correctness:** Yes, the dataset is in general constr…
**Clarity:** The paper is clearly-structured and w…

**Strengths:**

1. The paper introduces the largest to date collection of datasets for monitoring the SDGs with machine learning, providing a standardized benchmark for various SDGs tasks, which is not previously achieved.

2. The datasets are not only applicable to research on SDGs, but also a good resource to train/test machine learning algorithms on real-world data. Especially, the cross-task, multi-modal data is very valuable.

3. The authors have put a great effort minimizing the potential ethical issues regarding the dataset, have provided a detailed discussion on the limitation of the dataset, and have pointed out the potential usages and pitfalls of the dataset. All these are helpful guiding the community to use the dataset properly.

**Weaknesses:**

I find no obvious weakness in the paper. There's one minor concern: it would be beneficial to briefly discuss the baseline methods in addition to describing them in Appendix E (e.g., when applicable, how's their performance compared to those reported in the original papers? Does this tell the advantage / extra information of the new dataset compared to existing ones?)



**Additional Feedback:**

Personally moving the table A7 (summary of SOTA methods on the benchmark tasks) to the main text would make the main paper more self-contained; though it is also understandable that describing such a large-scale dataset within the page limit is challenging.

**Documentation:**

The supplementary material provides detailed documentations on how the data are collected, how the train/test splits are created, as well as the baseline methods.
The hosting, licensing and maintenance plan are also provided.

**Ethics:**

No. Although there are potential ethical concerns regarding privacy in some of the introduced datasets, the authors make an effort to remove or blur the information that may reveal private information.
The authors also provide (in Appendix F) how these are done, together with a detailed discussion in the Appendix, which is compelling.

**Relation To Prior Work:**

Yes, the paper clearly analyzes the limitations of prior work and states the strengths of this work against existing ones.

**Summary And Contributions:**

The paper provides a compilation of datasets for 15 benchmark tasks across 7 United Nation Sustainable Development Goals (SDGs). As the efforts towards the SDGs are highly interdisciplinary, they are less well accessible to the machine learning community. This paper is an effort to bridge the gap: it not only collects large-scale, cross-domain raw data for different SDG tasks, but also process them into a machine-learning-ready state, thereby creating a standardized resource that enables the machine learning researchers to contribute to these societal goals.

Contributions:
- the largest compilation to date of datasets for SDG monitoring that offers standardized, machine-learning ready, data splits for benchmarking.
- well-documented baseline models/results, and a public leaderboard that encourage machine learning community to contribute to the SDG problems.

---

> ### Author Response · Authors · 2021-09-30
> **Response to Reviewer uRQg**
>
> Thank you for your positive feedback and recognition of the contributions of the SustainBench dataset, as well as comments for how to improve the paper. Please find our response to your comments below. We also added a [top-level comment](https://openreview.net/forum?id=5HR3vCylqD&noteId=KLX4Yr4iaoL) to summarize the major changes made in response to all four reviewers’ comments. We have updated the manuscript PDF, and the Supplementary Material now includes a “tracked changes” PDF to help reviewers identify changes.
>
> _**Q:** More context on baseline models_
>
> **A:** For each specific task, we have expanded the relevant appendix sections to include more comparisons of the baseline models against related models in the literature. For example, we added a new, higher-performing baseline for field delineation based on the latest literature and described it in Appendix E.2.5.
>
> “While the original paper that compiled SustainBench's field delineation dataset achieved a Dice score of 0.61 with a standard U-Net [1], we applied a new attention-based CNN developed specifically for field delineation [2] and achieved a 0.87 Dice score. To our knowledge, this is the state-of-the-art deep learning model for field delineation.”
>
> In addition, in Appendix E.1, we have added more text to highlight the importance of including both satellite and street-level images for the DHS-based tasks. We (and others [4]) have observed that relative to predicting poverty, models trained only on satellite images are less predictive of child mortality rate, women BMI, women education, water index, and sanitation index. As there are no existing models that take advantage of both satellite and street-level imagery, we encourage researchers to develop novel algorithms that can utilize both input sources.
>
> However, in many cases it is quite challenging to compare our baselines against other models in the literature because of the lack of standardized datasets, train/test splits, and evaluation metrics. Indeed, a primary goal of SustainBench is to provide a standardized suite of benchmarks, which until now has been severely lacking.
>
> _**Q:** Move benchmarks table to main text_
>
> **A:** We have moved the benchmark table into the main text of the updated manuscript. It is now called Table 2. Note that we have been given an extra page in the revision (per email from program chairs: “you are allowed to upload an updated version of your manuscript and add an additional (10th) page”).
>
> **References**
>
> [1] H. L. Aung, B. Uzkent, M. Burke, D. Lobell, and S. Ermon. Farm parcel delineation using spatio-temporal convolutional networks. In Proceedings of the IEEE/CVF Conference on Computer Vision and Pattern Recognition Workshop, 2020.
>
> [2] ​​F. Waldner, F. I. Diakogiannis, K. Batchelor, M. Ciccotosto-Camp, E. Cooper-Williams, C. Herrmann, G. Mata, and A. Toovey. Detect, consolidate, delineate: Scalable mapping of field boundaries using satellite images. Remote Sensing, 2021.

---

### Official Review · Reviewer_6Wyr · 2021-09-21
**Important problem and a major contribution, but somewhat incomplete**

**Rating:** 5
**Confidence:** 3
**Clarity:** The paper is relatively well-written.

**Strengths:**

1) Through this effort, the researchers are connecting a large community of non-ML researchers (e.g., in public policy, economics, and energy sector) with the ML research community.  The website can ultimately provide a central location for non-ML researchers to see the state/progress of ML-based tools for monitoring SDGs.   Historically efforts to develop ML-based tools for SDGs have been largely uncoordinated, and difficult to navigate by ML researchers and non-ML researchers alike.
2) The authors release several new large datasets with labels.
3) More importantly, the authors put all of these datasets in one place, and make them easily accessible via Python code.  I agree with authors that a major challenge for ML researchers is accessing and pre-processing these large datasets.  Having the datasets available via download packages is really helpful.


**Weaknesses:**

1) The relationship to existing work is somewhat weak (see comments on "Relation To Prior Work") below.

2) Related to (1), the number of benchmark methods included seems to be limited, despite the fact that there are (I believe, though could be incorrect) substantial existing work on some of these problems.

3) The website appears to be somewhat incomplete. Some specific instances include the following:
a) The "Get Started" page has just a single sentence on it (https://sustainlab-group.github.io/sustainbench/docs/get_started.html)
b) Some of the links to download the datasets do not work, or are empty.  I refer to some specific examples under my response for the "Correctness" enquiry.
c) There are no benchmark results listed.







**Additional Feedback:**

No additional feedback.

**Correctness:**

Some of the links do download the datasets do not work or are empty, including those on the following pages of their website:
1) https://sustainlab-group.github.io/sustainbench/docs/datasets/sdg15/out_of_domain_land_cover.html
2) https://sustainlab-group.github.io/sustainbench/docs/datasets/sdg15/land_cover_representation.html
3) https://drive.google.com/drive/folders/1VvDQHTorD8sa6YJ6_Z9UoEFGu7QpR2dT
4) https://sustainlab-group.github.io/sustainbench/docs/datasets/sdg2/TBD

**Documentation:**

The needed documentation on data licenses, maintenance, etc. all seem to be present, either in the main manuscript or in the supplement.

**Ethics:**

Privacy is a major ethical consideration when applying recognition models to remote sensing data, where private property is often visible and may even be the target of the recognition models.  However, the authors mention this concern in Section 5 of the main paper, and provide a detailed discussion of these concerns, and their mitigation strategies, in the supplement.

**Relation To Prior Work:**

The work seems to be incomplete when comparing to prior work.  For many of the included benchmark tasks, I believe (though I'm not sure in all cases) that there are some other existing resources (e.g., datasets) and existing methods (e.g., models).  When proposing a new benchmark, it is important that the authors explain how their proposed dataset and scoring methods compare to existing work, and how their particular dataset is advantageous or differs compared to existing resources.  Similarly, while the authors often provide some justification for their chosen baseline methods (e.g., it is simple, or it comes from an existing paper), they often don't seem to discuss other methods on the topic, or how their chosen baseline compares to them.

For example, in D.3 could the authors clarify how their crop datasets compare to other crop datasets utilized in recent studies (e.g., in geographic coverage, number of classes, etc) and whether these prior works use public or private datasets?  Similarly, in E.2.2, E.2.3, and E.2.4 it would be helpful if the authors could explain how their proposed baseline methods for these problems compare to work by other authors on these topics?    I'm only somewhat familiar with this literature and so I could be mistaken, in which case the authors need only clarify that there are not existing methods/datasets beyond those they mention in their supplement.





**Summary And Contributions:**

The authors propose 15 benchmark tasks for inferring indicators of human/economic activity using (mostly) remote sensing data.  The benchmark tasks are quite unique in their explanatory (input) data and scoring metrics, however, they are assembled together in this benchmark because they are each useful for measuring progress towards one (or more) United Nations Sustainable Development Goal (SDG).

I believe this work addresses an important open problem.  At present it is difficult for ML-focused researchers to understand and contribute to SDGs; and it is similarly difficult for non-ML researchers to navigate the scattered research involving the use of ML methods to monitor SDGs.  This benchmark and the attendant resources provided by the authors (e.g., large newly-released datasets, code for accessing these data) are a keystone resource to connect and coordinate the efforts of these communities.

I would give a higher rating but the work seems somewhat incomplete.  The website seems incomplete, and I'm concerned by the limited inclusion/discussion of related models to solve some of the proposed benchmark tasks.  Similarly, I think further discussion of, and comparison with, related datasets for the proposed benchmark tasks is needed.   I explain these concerns further in the "Relation to Prior Work" section.

This is an ambitious project and I appreciate the substantial work that has gone into this already.  I think it is a great challenge to prepare all of the necessary content to support this project.  However, I still think everything must be complete and thorough for publication.   With that in mind, my rating would be much higher if some of the aforementioned limitations were addressed.

Also, this paper already contains a substantial amount of content, much of which is in a long supplement.  I think this project might be better-served by a longer-form journal publication, allowing for more explanation and detail in the main text.

---

> ### Author Response · Authors · 2021-09-30
> **References for Response to Reviewer 6Wyr**
>
> **References:**
>
> [1] ​​F. Waldner, F. I. Diakogiannis, K. Batchelor, M. Ciccotosto-Camp, E. Cooper-Williams, C. Herrmann, G. Mata, and A. Toovey. Detect, consolidate, delineate: Scalable mapping of field boundaries using satellite images. Remote Sensing, 2021.
>
> [2] CSIRO. ePaddocks Australian Paddock Boundaries. URL: https://acds.csiro.au/epaddock-australian-paddock-boundaries
>
> [3] L. Yan and D. Roy. Conterminous United States crop field size quantification from multi-temporal Landsat data.Remote Sensing of Environment, 2016.
>
> [4] National Agricultural Statistics Service. USDA National Agricultural Statistics Service Cropland Data Layer. Published crop-specific data layer [Online], 2018. URL https://nassgeodata.gmu.edu/CropScape/.
>
> [5] J. Inglada, M. Arias, B. Tardy, O. Hagolle, S. Valero, D. Morin, G. Dedieu, G. Sepulcre,S. Bontemps, P. Defourny, and B. Koetz. Assessment of an operational system for crop type map production using high temporal and spatial resolution satellite optical imagery. Remote Sensing, 2015.
>
> [6] R. Rustowicz, R. Cheong, L. Wang, S. Ermon, M. Burke, and D. Lobell. Semantic segmentation of crop type in Africa: A novel dataset and analysis of deep learning methods. In Proceedings of the IEEE/CVF Conference on Computer Vision and Pattern Recognition (CVPR) Workshops, 2019.
>
> [7] S. Wang, S. Di Tommaso, J. Faulkner, T. Friedel, A. Kennepohl, R. Strey, and D. B. Lobell. Mapping crop types in Southeast India with smartphone crowdsourcing and deep learning. Remote Sensing, 12(18), 2020.
>
> [8] H. Kerner, C. Nakalembe, and I. Becker-Reshef. Field-level crop type classification with k nearest neighbors: A baseline for a new Kenya smallholder dataset, 2020.
>
> [9] H. Zhao, S. Duan, J. Liu, L. Sun, and L. Reymondin. Evaluation of five deep learning models for crop type mapping using Sentinel-2 time series images with missing information. Remote Sensing, 2021.

---

> > ### Comment · Reviewer_6Wyr · 2021-10-06
> > **Thorough revisions, and most major concerns addressed**
> >
> > I thank the authors for their clear responses, and thorough revisions to the paper and the website.  Most of my major concerns have been addressed, or at least mitigated.  I am comfortable with acceptance of this paper in its latest form.

---

> > > ### Author Response · Authors · 2021-10-06
> > > **Thank you for your response**
> > >
> > > Thank you very much for considering our revisions and updating your evaluation of our paper. To state it more explicitly for the area chair, would you consider updating your numerical score in a comment?

---

> > > > ### Comment · Reviewer_6Wyr · 2021-10-07
> > > > **No longer able to update rating**
> > > >
> > > > Absolutely - I did attempt to update my rating but unfortunately I have been informed that we can no longer change our ratings.  However, I have informed the AC that I would like to improve my rating.

---

> ### Author Response · Authors · 2021-09-30
> **Response to Reviewer 6Wyr**
>
> We are grateful to this reviewer for their very detailed feedback, which has helped to greatly improve our work. Please find our point-by-point response to the reviewer’s comments below. We also added a [top-level comment](https://openreview.net/forum?id=5HR3vCylqD&noteId=KLX4Yr4iaoL) to summarize the major changes made in response to all four reviewers’ comments. We have updated the manuscript PDF. The Supplementary Material now includes a “tracked changes” PDF to help reviewers identify changes.
>
> _**Q:** The website appears to be somewhat incomplete._
>
> **A:** Thank you for checking the pages and links on our website and pointing out ones that are underdeveloped or do not work. We have updated our website to include an accurate leaderboard, a getting started page with code examples, and corrected links to datasets.
>
> _**Q:** Explain how the proposed datasets and scoring metrics compare to existing work._
>
> **A:** We have added a new table (Table 1) on the most relevant dataset comparisons to the main text. For each specific task, we have expanded the relevant appendix sections to compare the included datasets against other similar datasets in the literature. For example, we added the following paragraph about our field delineation dataset:
>
> “To our knowledge, SustainBench has released the first public field boundary dataset with satellite image inputs and ML-friendly outputs. That is, some countries in Europe (e.g. France) have made vector files of field boundaries public on their government websites, but without corresponding satellite imagery inputs or raster field boundary outputs. We provide these inputs and outputs. While field segmentation datasets from the US, South Africa, and Australia were used in prior field delineation research [1,2,3], none of those datasets are publicly available. We are also currently working on collecting field boundaries in low-income countries, but this data will be added to SustainBench at a later date, not in time for this submission.”
>
> For the crop type datasets in Ghana, South Sudan, and Kenya, we added Table A8 in the appendix to compare existing public datasets with the two datasets in SustainBench. Please see the updated manuscript for all of the updates.
>
> _**Q:** Compare methods from prior work to the paper’s chosen baseline models._
>
> **A:** We have updated the relevant appendix sections to include more comparisons of the baseline models against related models in the literature. For example, we have added the following paragraph to Appendix Section E2.2.
>
> “Like cropland maps, most operational works classifying crop types employ SVM or random forest classifiers [4,5]. The baseline model that we use from [6] improves upon these by using an LSTM-CNN. Recent models used in other, non-operational works include 1D CNNs and 3D CNNs [7] and kNN [8]. A review from this year comparing five deep learning models found that 1D CNN, LSTM-CNN, and GRU-CNN all achieved high accuracy on classifying crop types in China, with differences between them statistically insignificant [9].”
>
> We did add a new baseline to the field delineation task, as it came to our attention that the FracTAL Res-UNet architecture from [7] significantly outperforms a standard U-Net for this task. This model is now in Table 2 and discussed in Appendix E.2.5.
>
> However, in general, it is quite challenging to compare our baselines against other models in the literature because of the lack of standardized datasets, train/test splits, and evaluation metrics. Indeed, a primary goal of SustainBench is to provide a standardized suite of benchmarks, which until now has been severely lacking.
>
> _**Q:** Consideration of journal publication (“Also, this paper already contains a substantial amount of content, much of which is in a long supplement. I think this project might be better-served by a longer-form journal publication, allowing for more explanation and detail in the main text.”)_
>
> **A:** Thank you for the suggestion. We agree that the large scope of SustainBench makes it challenging to describe dataset details within 10 pages of main text, and we will consider a follow-up journal publication. However, our goal is to specifically reach out to the ML community, and we feel that NeurIPS and this track in particular best serves that goal. We hope that the concise dataset summaries in Section 3, along with the addition of Tables 1 and 2 in this revision, can help ML researchers quickly see multiple options for where their skills can contribute to SDG monitoring. Then those who are interested in a particular SDG or dataset can refer to the Appendix for more details.

---

### Official Review · Reviewer_jAns · 2021-09-21
**Important area to address, could use more work**

**Rating:** 6
**Confidence:** 2
**Clarity:** Paper is clearly written and well pre…

**Strengths:**

- UN’s sustainable development goals is a crucial area where machine learning community can help in monitoring if there are standardized benchmarks and SUSTAINBENCH tries to address this issue.
- Many different countries are covered in the dataset across different tasks and some of the data is gathered from different previous work. While this makes the data a bit scattered, it is a good start to collect them in one place for a common goal.

**Weaknesses:**

- For some tasks that uses DHS surveys like predicting Women BMI, women education and child mortality rate; it is not clear how using satellite images as in input can help predicting women education levels at all. Low r2 numbers on the baseline model shows this too (i.e. 0.01). And these are the only evaluation results that do not rely on previously published work.
- It’s not clear how some of the specific tasks defined helps with improving the SDG. For example, it’s not clear to me how does field delineation task helps with Zero Hunger SDG, it would have been good to provide more background and motivation on this for non domain experts.
- While predicting poverty over space task includes both satellite and street level images, predicting poverty over time only contains satellite images. Since these two tasks are related, it would have been more complete if predicting poverty over time task included street level images over time as well. Figure 1 makes it look like it does include but in fact it does not.
- The leaderboard on the website does not contain any results/baselines. It would have been great if Table A7 included the model type used to evaluate as well.



**Additional Feedback:**

Overall I think it is a crucial area and I think the SustainBench could be valuable benchmark suite. Please address the questions in the weaknesses section.

**Correctness:**

The evaluation results in the paper are collected/gathered mostly from previously published peer-reviewed results. I added some comments about the ones that are not priorly published above.

**Documentation:**

Yes.

**Ethics:**

No.

**Relation To Prior Work:**

Relation to prior work is discussed. SustainBench build on many prior work that focuses on one particular SDG related task, SustainBench aims to collect many tasks towards different SDGs and standardize towards a collective goal. It also adds new data which was not used in prior work before.

**Summary And Contributions:**

The paper releases a collection of benchmarks and datasets related to UN’s sustainable development goals (SGD). Monitoring SDGs is challenging since it requires ground based household surveys and domain knowledge. SUSTAINBENCH aims to define standard datasets and metrics to make prediction of SDG tasks using machine learning easier.

---

> ### Author Response · Authors · 2021-09-30
> **Response to Reviewer jAns**
>
> Thank you for your thoughtful feedback, recognition of SustainBench’s contributions, and suggestions for improvement. We provide point-by-point responses below.
>
> We added a [top-level comment](https://openreview.net/forum?id=5HR3vCylqD&noteId=KLX4Yr4iaoL) to summarize major changes made in response to all reviewers’ comments. We updated the manuscript PDF; the Supplementary Material includes a “tracked changes” PDF to help reviewers identify changes.
>
> _**Q:** Unclear how satellite images can predict some DHS-based labels (e.g., women BMI, women education, child mortality). These evaluations do not rely on previously published work._
>
> **A:** We updated Section 3 (Datasets) and Appendix to explain why we believe DHS-based labels can be predicted from imagery. We also added more context for the implications of the low $r^2$ numbers from the baseline model. A summary of our additions is below.
>
> For 4 of the 6 DHS-based tasks, prior works show that satellite and/or street-level imagery are predictive inputs. However, these works use different labels from SustainBench, making it difficult to compare results.
>
> | | Satellite image | Street-level input |
> |-|-|-|
> | poverty | Yeh 2020, Jean 2016 [1,2] | Lee 2021, Gebru 2017 [3,4] |
> | women BMI | Head 2017, Maharana 2018 [8, 5] | Lee 2021 [3] |
> | women education | Head 2017, Zhao 2020 [8, 6] | Gebru 2017 [4] |
> | clean water | Head 2017 [8] | none |
>
> (This is a truncated version of the newly added Table A5 in the Appendix.)
>
> For the other 2 tasks without prior works using imagery inputs:
> * Sanitation index: the survey variable for the sanitation index is also used in the asset wealth index. Since the asset wealth index can be estimated from satellite imagery, it is reasonable to expect that the sanitation index is also predictable. Indeed, the baseline model is able to explain 36% (sanitation index) of the variation in the labels from nightlights alone.
> * Child mortality: While child mortality is difficult to directly observe from imagery, such imagery can capture environmental factors that are highly correlated with child mortality. Child health is strongly related to nutritional status, which is linked to agricultural conditions that are visible in satellite imagery. Based on a study of child mortality in Ghana, “there is evidence that child mortality is connected to environmental factors such as housing quality, slum-like conditions, and neighborhood levels of vegetation” [7].
>
> Note that our baseline model for DHS surveys only uses the nightlights (NL) band from satellite imagery. Thus, the main conclusion from our baseline results is that NL is predictive of poverty, but less so for the other DHS-based labels (women BMI, child mortality, women edu, water quality, and sanitation). This is a useful finding, because it suggests that predicting these other labels likely requires different models and/or inputs. We draw attention to 1) the need to develop better models for predicting these labels, and 2) the availability of both satellite & street-level imagery for these tasks.
>
> Finally, if other researchers find better data sources for predicting these SDG indicators, we note in our paper that they are welcome to use our labels and submit results to our leaderboard. (Such results would be tracked separately.)
>
> _**Q:** More clarity for how some tasks help with improving SDGs. E.g., how does field delineation help with Zero Hunger? More background + motivation for non domain experts._
>
> **A:**  We updated our paper to clarify how each task in SustainBench helps monitor progress towards the SDGs and provide context for a general audience.
>
> Specifically for field delineation, we added the following paragraph to Section 3.2.4 to explain how it helps achieve SDG 2, along with 5 new references that used field boundary data to study productivity, mechanization, species richness, etc.
>
> “Since agricultural practices are usually implemented on the level of an entire field, field boundaries can help reduce noise and improve performance when mapping crop types, yields, and management decisions. Furthermore, field boundaries are a prerequisite for today's digital agriculture services that help farmers optimize yields and profits. Statistics that can be derived from field delineation, such as the size and distribution of crop fields, have also been used to study productivity, mechanization, and biodiversity. Field boundary datasets are rare and only sparsely labeled in low-income regions, so we release a large dataset from France to aid in model development.”
>
> _**Q:** Incorporate street-level images for task of predicting changes in poverty over time._
>
> **A:** We wanted to include street-level images, but only 16 (1%) of the 1287 clusters in this task had matching street-level images from Mapillary. We do not think it is useful to include such few images. We updated the paper to highlight this.
>
> _**Q:** Add results/baselines to leaderboard. Add model type to Table A7._
>
> **A:** Fixed.

---

> ### Author Response · Authors · 2021-09-30
> **References accompanying our response to Reviewer jAns**
>
> **References:**
>
> [1] C. Yeh, A. Perez, A. Driscoll, G. Azzari, Z. Tang, D. Lobell, S. Ermon, and M. Burke. Using publicly available satellite imagery and deep learning to understand economic well-being in Africa. Nature Communications, 2020.
>
> [2] N. Jean, M. Burke, M. Xie, W. M. Davis, D. B. Lobell, and S. Ermon. Combining satellite imagery and machine learning to predict poverty. Science, 2016.
>
> [3] ​​J.  Lee,  D.  Grosz,  B.  Uzkent,  S.  Zeng,  M.  Burke,  D.  Lobell,  and  S.  Ermon.   Predicting Livelihood Indicators from Community-Generated Street-Level Imagery. Proceedings of the AAAI Conference on Artificial Intelligence, 2021.
>
> [4] ​​T. Gebru, J. Krause, Y. Wang, D. Chen, J. Deng, E. L. Aiden, and L. Fei-Fei. Using deep learning  and  Google  Street  View  to  estimate  the  demographic  makeup  of  neighborhoods across the United States. Proceedings of the National Academy of Sciences, 2017.
>
> [5] A. Maharana and E. O. Nsoesie. Use of Deep Learning to Examine the Association of the Built Environment With Prevalence of Neighborhood Adult Obesity. JAMA Network Open, 2018.
>
> [6]  S. Zhao, C. Yeh, and S. Ermon. A Framework for Sample Efficient Interval Estimation with Control Variates. In Proceedings of the 23rd International Conference on Artificial Intelligence and Statistics, 2020.
>
> [7] M. M. Jankowska, M. Benza, and J. R. Weeks. Estimating spatial inequalities of urban child mortality. Demographic research, 2013
>
> [8] A. Head, M. Manguin, N. Tran, and J. E. Blumenstock. Can Human Development be Measured with Satellite Imagery? In Proceedings of the Ninth International Conference on Information and Communication Technologies and Development, pages 1–11, Lahore, Pakistan, 11 2017.526ACM. ISBN 978-1-4503-5277-2. doi: 10.1145/3136560.3136576.

---

### Author Response · Authors · 2021-09-30
**Overall Author Response to Reviews**

We are grateful to all the reviewers for their constructive feedback. We appreciate that all of the reviewers recognize the importance of the UN SDGs and the possibility of using ML methods to help measure progress towards the sustainability goals. As Reviewers jAns, 6Wyr, and uRQg highlighted, the key contribution of SustainBench is that it is the largest to-date collection of standardized, pre-processed, and easily accessible datasets specifically designed for SDG monitoring with clearly defined train/val/test splits for machine learning benchmarking. We are also glad that Reviewer uRQg recognizes SustainBench as a valuable dataset for testing machine learning algorithms on real-world data, especially for multi-task, multi-modal settings.

The main constructive criticism given by the four reviewers was to elaborate on SustainBench's real-world impact, more thoroughly compare SustainBench datasets and baseline models to existing works, and improve our website. In response, we have revised the manuscript extensively. We have:
1. added examples to _Section 1: Introduction_ and _Section 5: Impact, Limitations, and Future Work_ of the manuscript to illustrate the impacts of SustainBench;
2. provided extensive new comparisons with related datasets and baseline models to highlight the novelty of SustainBench (see the new Table 1, Sections 2-4, and Appendix); and
3. updated our [website](https://sustainlab-group.github.io/sustainbench/) with a populated [leaderboard](https://sustainlab-group.github.io/sustainbench/leaderboard/), [code samples](https://sustainlab-group.github.io/sustainbench/docs/get_started.html#data), and [corrected dataset links](https://sustainlab-group.github.io/sustainbench/docs/datasets/).

**We emphasize that SustainBench has the potential to impact both the machine learning community and global society.** For many tasks that have been previously studied in the literature (such as poverty prediction, crop yield forecasting, and landcover mapping), the lack of standardized benchmarks has limited our ability to measure modelling performance on these tasks; SustainBench begins to fill this major gap. For some other tasks (such as predicting child mortality and sanitation), SustainBench is the first dataset to introduce them to the ML community (to the best of our knowledge). Just as ImageNet catalyzed ML research in object recognition, we hope that SustainBench will spur novel algorithmic development for models that can take advantage of diverse, real-world sustainability-related datasets.

We'd also like to address the rationale of releasing such a large dataset suite, made up of many individual datasets. We believe there is significant value in having all of these datasets in one place for researchers to easily see what modeling problems are solved vs. where there is room for progress. This enables the ML and sustainability research communities to allocate their time and effort accordingly. In addition, as we explicitly stressed in the manuscript, many sustainability indicators are highly correlated, and we want researchers to be able to take advantage of the synergy between these datasets.

We have updated the manuscript PDF, which is now 10 pages long (compared to originally 9), as allowed for the revision. Additionally, we have included a “tracked changes” version of the manuscript inside the Supplementary Material zip file to help reviewers see where changes were made (blue text = new text).

---

### Decision · Program_Chairs · 2021-10-09

**Decision:**

Accept

**Comment:**

Dear authors,

Thank you for submitting your paper and addressing the issues raised by the reviewers.

The paper proposes a set of 15 problems for monitoring Sustainable Development Goals (SDGs) with machine learning. They provide a python module with dataloaders similar to WILDS and a website with leaderboards for each dataset.

The main criticism was

Missing comparison with existing work Incompleteness of the website Some tasks were not described clearly

All three points were addressed by the authors in the rebuttal. Furthermore, this work allows the ML community easier access to data related to the highly important sustainable development goals.

Taking all this into account as well as the scores of 6,5,7,7 the paper is accepted.